# Microstructural Dynamics of Polymer Melts during Stretching: Radial Size Distribution

**DOI:** 10.3390/polym15092067

**Published:** 2023-04-26

**Authors:** Ming-Chang Hsieh, Yu-Hao Tsao, Yu-Jane Sheng, Heng-Kwong Tsao

**Affiliations:** 1Department of Chemical Engineering, National Taiwan University, Taipei 106, Taiwan; hsiehmingchang311@gmail.com (M.-C.H.); simon84111@gmail.com (Y.-H.T.); 2Department of Chemical and Materials Engineering, National Central University, Jhongli 320, Taiwan

**Keywords:** microstructural dynamics, elongational viscosity, strain hardening, radial size distribution, dissipative particle dynamics

## Abstract

The transient elongational viscosity $\eta_{e}\left(t\right)$ of the polymer melt is known to exhibit strain hardening, which depends on the strain rate $\dot{\varepsilon }$. This phenomenon was elucidated by the difference of chain stretching in the entanglement network between extension and shear. However, to date, the microscopic evolution of polymer melt has not been fully statistically analyzed. In this work, the radial size distributions *P*($R_{g},t$) of linear polymers are explored by dissipative particle dynamics during the stretching processes. In uniaxial extensional flow, it is observed that the mean radius of gyration ${\bar{R}}_{g}\left(t\right)$ and standard deviation σ(t) remain unchanged until the onset of strain hardening, corresponding to linear viscoelasticity. Both ${\bar{R}}_{g}$ and σ rise rapidly in the non-linear regime, and bimodal size distribution can emerge. Moreover, the onset of strain hardening is found to be insensitive to the Hencky strain (${\dot{\varepsilon }}_{H}t$) and chain length (N).

## 1. Introduction

The mechanical responses to the extension of entangled polymer melts are crucial in rheology for guiding polymer processing [1]. For example, in the processes of film manufacturing, fiber spinning, and blow molding, extensional deformation is the dominant mode. Moreover, as the melt flows through a sudden contraction, it accelerates, and the extensional flow becomes predominant. Since those processes are often conducted under highly non-linear conditions, the extensional rheology cannot be derived from the shear rheological behavior. The extensional viscosity describing the resistance of a fluid to the extensional flow is frequently used to characterize the molten polymer, which is viscoelastic. The elongational viscosity ${(\eta }_{e})$ associated with the uniaxial extension is defined as the ratio of normal stress difference (τ) to the rate of strain ($\dot{\varepsilon }$), and the ratio between the elongation and shear viscosities, known as Trouton’s ratio, is equal to three for Newtonian fluids.

A lot of experimental set-ups have been developed for measuring extensional viscosity, such as Göttfert rheotens tester [2] and CaBER [3,4,5,6,7]. However, it is still a challenge to acquire $\eta_{e}$ accurately. The factors limiting the improved characterization of $\eta_{e}$ include flow instabilities and the rupture of the sample [8]. Moreover, spatial uniformity is difficult to achieve in producing purely uniaxial extension, and the steady-state elongation flow is seldom reached [9]. Measurements at a constant elongational rate ($\dot{\varepsilon }$) corresponding to stressing experiments are easier to perform. As a result, the stress has to increase with time τ(t) in order to maintain constant $\dot{\varepsilon }$ in uniaxial extensional flow, leading to the transient elongational viscosity $\eta_{e}\left(t\right)$. It is known that the elongational viscosity ascends with the time of deformation and follows the linear viscoelastic response in the zero-strain rate limit, $\eta_{0}\left(t\right)$. However, at finite strain rates, an exciting feature of the elongational flow occurs, named strain hardening. During the stretching process, the non-linear response emerges, and the transient extensional viscosity starts to deviate upward from $\eta_{0}\left(t\right)$, recognized as the onset of strain hardening [1,10]. The behavior of $\eta_{e}\left(t\right)$ has been reported to be greatly affected by the strain rate, which is mainly manifested in strain hardening. As $\dot{\varepsilon }$ is increased in stressing experiments, the onset of strain hardening takes place in a shorter stretching time. That is, the non-linear viscoelasticity appears earlier.

It was thought that strain hardening was a feature indicative of specific polymer architecture and closely associated with long-chain branching [11]. However, even $\eta_{e}\left(t\right)$ of fairly monodisperse melts was found to display an upward deviation at sufficiently high strain rates [8]. In elongational flows, chain stretching is commonly believed to be responsible for strain hardening. Recently, the origin of strain hardening has been elucidated based on the comparison of the responses between uniaxial extension and simple shear [1]. Regardless of polymer architectures, well-entangled polymer melts tend to exhibit strain hardening because the chain disentanglement effect is partially suppressed in extension. Moreover, the geometric condensation of the load-bearing chains accompanied by the shrinking cross-sectional area strengthens the entanglement network effectively [1]. In general, the effect of the strain rate on the onset of strain hardening is simply manifested by the onset time. Nonetheless, some experimental studies from a few decades ago reported that the onset of the rapid stress growth took place at an approximately constant value of the total Hencky strain [12,13].

The dynamics of entangled polymer melts have been extensively studied by the reptation (or tube) model [14] and the molecular stress function (MSF) model [15,16]. In the former, the dynamical behavior of a chain due to its interactions with surrounding chains is accounted for by the imposed topological constraints. To estimate $\eta_{e}$ quantitatively, the MSF model is introduced by incorporating the interchain tube pressure effect into the reptation model, and it contains certain input parameters that are typically acquired from experiments [16,17]. Indeed, the molecular mechanism responsible for the elongational behavior of entangled polymers can be scrutinized via microscopic simulations, such as the primitive chain network (PCN) model [18,19] and molecular dynamics (MD) simulation [20,21]. In the PCN model, the rubber-like network is constructed by a sequence of subchains connecting consecutive entanglements (nodes) [18]. Network rearrangement in real space is facilitated by various chain dynamical mechanisms, including reptation, contour length fluctuation, and constraint release, based on the force balance on entanglements [19]. However, to agree with the experiments, appropriate molecular mechanisms such as stretch/orientation-dependent monomeric friction [18] have to be incorporated into the PCN model. In contrast, MD can provide direct information about chain dynamics in polymer processing [22,23] without prior knowledge of molecular mechanisms. Nonetheless, the high computational costs make MD difficult to simulate larger systems for longer time scales. As a result, the MD approach is seldom employed to study the elongational flow of polymer melt [20].

Coarse-grained molecular dynamics is able to offer microscopic information about stress, chain conformation, and dynamics [24] while requiring a more reasonable computational cost. Complications in experiments, including molecular weight distribution, can be eliminated simply in simulations. By representing a polymer as beads and springs, the steady-state elongational viscosity of weakly entangled polymer melt has been observed after strain hardening [25]. By mimicking the behavior of extensional flows in entangled polymer melts, observed patterns in extensional viscosity regarding time, rate, and molecular weight are replicated and chain conformations are related to extensional stress [26]. Recently, the viscosity overshoot of a blend of the ring and linear polymers has been reported for biaxial extensional flows [27]. In addition, in this paper, a new simulation methodology that uses a hierarchical triple scale approach is demonstrated to predict the dynamic and rheological characteristics of entangled polymer melts with a high molecular weight [28]. However, until now, the microstructural dynamics of polymer melts during stretching have not been analyzed directly. Particularly, polymer conformations associated with strain hardening have not been microscopically examined. In this work, the elongational flow of linear polymer melts is explored by coarse-grained dissipative particle dynamics (DPD). Both weakly and strongly entangled polymer melts are considered by varying the chain length. The transient elongational viscosity is acquired for various strain rates, and strain hardening is identified. The microstructural dynamics are investigated by monitoring the evolution of the radial size distribution of polymers. The relationship between strain hardening and the mean radius of gyration (and standard deviation) is then established. The influences of the strain rate and molecular weight on the onset of strain hardening are studied.

## 2. Methods

Similar to coarse-grained MD, DPD is a particle-based mesoscale simulation method [29] that can be used to investigate larger length and time scales than conventional molecular dynamics (MD) [29,30,31]. The DPD bead with mass *m* is comprised of a few molecules or atoms, and the time evolution of the beads is governed by Newton’s equation of motion [32,33]. The forces acting on the DPD bead are generally classified into three types: conservative force ($f_{\mathrm{ij}}^{C}$), dissipative force ($f_{\mathrm{ij}}^{D}$), and random force ($f_{\mathrm{ij}}^{R}$). These forces are soft and repulsive, in addition to being short-ranged and pairwise-additive [33]. As the distance between any two beads ($r_{\mathrm{ij}}$) is larger than the cut-off radius ($r_{c}$), these forces vanish. The total force exerting on bead i is then $F_{i}=\sum_{j(\neq i)}{(f}_{\mathrm{ij}}^{C}{+f}_{\mathrm{ij}}^{D}{+f}_{\mathrm{ij}}^{R})$. The conservative force decreases linearly with interparticle distance, $f_{\mathrm{ij}}^{C}=a_{\mathrm{ij}}(1-r_{\mathrm{ij}}{/r}_{c}{)\hat{r}}_{\mathrm{ij}}$, where the interaction parameter $a_{\mathrm{ij}}$ denotes the interaction strength between two beads and ${\hat{r}}_{\mathrm{ij}}$ is the unit vector of the inter-particle distance. Because the polymer melt is made of the same type of DPD bead, the interaction parameter between all pairs of beads is always set as $a_{\mathrm{mm}}=25$ [34]. $f_{\mathrm{ij}}^{D}$ is proportional to the relative velocity between two DPD beads and $f_{\mathrm{ij}}^{R}$ is introduced to satisfy the fluctuation–dissipation theorem [35,36,37,38]. The conservation of momentum in DPD simulations is automatically fulfilled by all pairwise interacting forces between every two DPD particles [31,39]. As a result, the hydrodynamic behavior of the system can be observed more easily compared to MD with too many details of the molecular motion [31]. All the units in our simulations are scaled by the bead mass (*m*), cut-off distance ($r_{c}$), and thermal energy ($k_{B}T$). Therefore, the time (t) is nondimensionalized by ${{(mr}_{c}^{2}{/k}_{B}T)}^{1/2}$ and stress ($\tau_{\mathrm{ij}}$) by $k_{B}{T/r}_{c}^{3}$. In this work, the open software Large-scale Atomic/Molecular Massively Parallel Simulator (LAMMPS) [40] was used for all simulations.

A linear polymer consists of repeating units linked only to two others [41]. It is made of a string of DPD beads, and each polymer chain possesses N beads. For simplicity, the molecular weight or chain length is represented by N. The string of DPD beads is connected with the harmonic spring (bond), $F_{\mathrm{ij}}^{S}=k_{s}{(r}_{\mathrm{ij}}-r_{\mathrm{eq}}{)\hat{r}}_{\mathrm{ij}}$, where the spring constant $k_{s}=100$ and equilibrium length $r_{\mathrm{eq}}=0.4$. Note that the original DPD parameters proposed by Groot and Warren [33] are too soft and provide no specific restrictions for steric interactions. Thus, chain crossings may occur. However, our choice of the DPD parameters ($r_{\mathrm{eq}}{,k}_{s}{,\;\mathrm{and}\;a}_{\mathrm{ij}}$) can effectively maintain the uncrossability of polymeric chains, and at the same time, the large integration step can still be employed. The verification of chain uncrossability is illustrated in the Appendix A. Appendix A also demonstrate that our choice of DPD parameters can indeed prevent chain-crossing events. The chain stiffness can be tuned by implementing additional bending forces between two consecutive bonds. The bending force, $F^{\theta }=-\nabla U^{\theta }$, is obtained by the bending potential, $U^{\theta }=k_{\theta }{(\theta -\theta_{\mathrm{eq}})}^{2}$. Here, the bending constant is $k_{\theta }=2,$ and the equilibrium angle is $\theta_{\mathrm{eq}}=\pi$. In this work, the number density of the linear polymer system is set as ρ=3. There are about ${6\times 10}^{5}$ DPD beads in the cuboid with a periodic boundary in all three spatial directions. Initially, the system is a cubic box with approximately 58.5×58.5×58.5. The time step is chosen as Δt=0.01 to avoid the divergence in simulations. Before conducting the elongation study, it is necessary for the simulation to reach equilibrium. At equilibrium, the properties of the system, such as the internal energy and radius of gyration, should remain constant over time.

In the uniaxial extension experiment, melts are stretched uniaxially at a constant strain rate ($\dot{\varepsilon }$) and then the extensional stress (τ) is measured. To maintain the constant deformation rate, the uniaxial stress has to be adjusted with time, leading to the time-varying elongational viscosity $\eta_{e}(t)=\tau (t)/\dot{\varepsilon }$. The extensional stress is ${\tau =\tau }_{\mathrm{xx}}-{(\tau }_{\mathrm{yy}}{+\tau }_{\mathrm{zz}})/2$ [42], where the average virial stress $\tau_{\alpha \alpha }$ is the negative value of the diagonal component $P_{\alpha \alpha }$ of pressure tensor in the α direction [43]. $P_{\alpha \alpha }$ can be computed by $P_{\alpha \alpha }=(\sum_{i}m^{i}v_{\alpha }^{i}v_{\alpha }^{i}/V)+0.5(\sum_{(j\neq i)}F_{\alpha }^{\mathrm{ij}}r_{\alpha }^{\mathrm{ij}}/V)$, where V is the volume of the simulation box. The first term relates to the components of the kinetic energy tensor, where $m^{i}$ and $v_{\alpha }^{i}$ represent the mass and α-component of the velocity of bead *i*, respectively. The second term uses the components of the virial tensor, where $F_{\alpha }^{\mathrm{ij}}$ and $r_{\alpha }^{\mathrm{ij}}$ denote the α-component of the force and distance between beads i and j, respectively [44]. It is calculated for all pairwise interactions. In this work, the simulation cubic box ($l_{0}{}^{3}$) is stretched at a constant “true” strain rate (i.e., Hencky strain rate, ${\dot{\varepsilon }}_{H}$) in the *x*-direction; thus, the *x*-dimension box size grows non-linearly with time, ${l(t)=l}_{0}{\exp(\dot{\varepsilon }}_{H}\Delta t)$. Note that it is different from the elongational process based on the constant “engineering” strain rate (${\dot{\varepsilon }}_{e}$). In order to keep the box volume constant, both of the box sizes in the *y*- and *z*-dimensions shrink equally over time.

The polymer melt has a lot of polymer conformations which can be simply described by the radial size distribution *P*($R_{g}$), where the radial size of a chain is represented by the radius of gyration ($R_{g}$). The polymer size along the *x*-axis is depicted by $R_{\mathrm{gxx}}=[\sum_{i}{{(x}_{i}-x_{\mathrm{cm}})}^{2}{/N]}^{1/2}$, where $x_{\mathrm{cm}}$ is the center-of-mass position. The radius of gyration of the polymer is $R_{g}={(R}_{\mathrm{gxx}}{}^{2}{+R}_{\mathrm{gyy}}{}^{2}{+R}_{\mathrm{gzz}}{}^{2}{)}^{1/2}$. The aspect ratio of the polymer is defined as ${s=R}_{\mathrm{gxx}}{/((R}_{\mathrm{gyy}}{+R}_{\mathrm{gzz}})/2)$. Certainly, the mean value of s at equilibrium is unity. During the stretching process, both the $R_{g}$ and s of each chain can vary with time and the radial size distribution of the polymer melt *P*($R_{g},t$) also changes as a function of time. The first moment of *P*($R_{g},t$) is ${\bar{R}}_{g}\left(t\right)$, which combines the second moment to give the standard deviation σ of the distribution.

## 3. Results

The elongational viscosity ($\eta_{e}$) is commonly determined by extensional flow experiments, such as filament-stretching via an extensional rheometer [45] and capillary break-up via an extensional rheometer [4,5,6,7]. In those experiments, melts are stretched uniaxially at a constant strain rate (${\dot{\varepsilon }}_{H}$), and then the extensional stress ${\tau =\tau }_{\mathrm{xx}}-{(\tau }_{\mathrm{yy}}{+\tau }_{\mathrm{zz}})/2$ is measured. To maintain the constant deformation rate, the uniaxial stress has to be adjusted with time, leading to the time-varying elongational viscosity $\eta_{e}{(t)=\tau (t)/\dot{\varepsilon }}_{H}$. The transient elongational viscosity depends on the strain rate $\eta_{e}{(t,\dot{\varepsilon }}_{H})$, and the characteristic stretching time corresponding to the onset of strain hardening decreases with increasing ${\dot{\varepsilon }}_{H}$ [1,46]. It is important to note that stretching and shearing are two distinct types of deformation processes, and the trends observed in our results for the stretching process may not necessarily apply to the shearing process.

### 3.1. Evolution of Radial Size Distribution of Polymers during Stretching

Figure 1a shows the simulation results of the time-varying elongational viscosities for linear polymers with N=130 subject to various strain rates. For the purpose of comparison, the elongational viscosity is non-dimensionalized by the shear viscosity of monomers (μ=0.85) [47]. For sufficiently low strain rates (e.g., ${\dot{\varepsilon }}_{H}={5\times 10}^{-4}{\;\mathrm{and}\;10}^{-3}$), their curves of $\eta_{e}\left(t\right)$ coincide with each other essentially. This consequence of the strain rate-independent behavior seems to imply that these two curves can represent the linear viscoelastic response $\eta_{0}{(t)=\eta }_{e}{(t,\dot{\varepsilon }}_{H}\to 0)$. Evidently, other curves of $\eta_{e}\left(t\right)$ at different ${\dot{\varepsilon }}_{H}$ (${\dot{\varepsilon }}_{H}\geq {5\times 10}^{-3}$) overlap with $\eta_{0}\left(t\right)$ initially but deviate at different stretching times, indicating the occurrence of strain hardening. The stress in the non-linear regime associated with strain hardening increases with time faster than that in the linear regime. Moreover, the onset point of strain hardening decreases with increasing strain rate. Obviously, the qualitative characteristics of the transient elongational viscosity acquired from DPD simulations agree well with those observed in elongational rheology experiments. The general explanation for the effect of the strain rate on strain hardening is that, as the strain rate is increased, the polymer chains will be stretched more rapidly and reach the non-linear viscoelastic regime earlier.

It is imperative to realize the strain rate effect in terms of the onset time (abscissa) of strain hardening. To further understand the influence of the strain rate on the elongational viscosity, the data of $\eta_{e}{(t,\dot{\varepsilon }}_{H})$ can be transformed into $\eta_{e}{(\varepsilon }_{H}{,\dot{\varepsilon }}_{H})$ based on the Hencky strain $\varepsilon_{H}={\ln(l/l}_{0})$, where l and $l_{0}$ denote the instantaneous and initial length of the sample [10]. That is, the transient elongational viscosity is plotted against the elongational deformation, as shown in the inset of Figure 1a. It can be observed that the elongational viscosity decreases with the strain rate subject to the same $\varepsilon_{H}$. The simulation outcome is qualitatively consistent with the experimental result [48,49]. It is known that, at very small strain rates, the polymer deformation is allowed to relax back to the shape close to equilibrium. As a result, the resistance to further deformation is weak, and the stress τ increases slowly. However, for larger strain rates, the relaxation process is relatively slow compared to the deformation rate [50]. Therefore, the nonequilibrium entanglement structure is strengthened to impede further deformation, leading to an increment of τ to maintain the constant strain rate. Although τ grows with ${\dot{\varepsilon }}_{H}$, their ratio $\eta_{e}{(\varepsilon }_{H}{,\dot{\varepsilon }}_{H}{)=\tau /\dot{\varepsilon }}_{H}$ actually decreases with increasing ${\dot{\varepsilon }}_{H}$ at a fixed $\varepsilon_{H}$, indicating that the increasing rate of τ is less than that of ${\dot{\varepsilon }}_{H}$. However, as the polymer melt is stretched to a specified length, more work is still required for higher strain rates based on the plot of τ versus $\varepsilon_{H}$.

Obviously, the steady state elongational viscosity, which is independent of time $\eta_{e}{(t)=\eta }^{*}$ [48], is not attained for N=130, as shown in Figure 1a. Presumably, a very large simulation box is required to sustain the long-time extension for reaching the steady state $\eta_{e}$ [25]. To demonstrate the possibility of obtaining $\eta^{*}$, the polymer melt with short chains N=10 is considered. Figure 1b shows the evolution of the elongational viscosity with time at different strain rates. The comparison between Figure 1a,b reveals significant differences between short and long chains. Although the initial values of $\eta_{e}$ of both N=10 and 130 have small differences, the latter becomes much greater than the former after stretching the polymer melts at the same ${\dot{\varepsilon }}_{H}$. This is not surprising because of the fewer entanglements in polymer melts and lower resistances to stretching for N=10. Moreover, for sufficiently low strain rates (${\dot{\varepsilon }}_{H}=10^{-2}{\;\mathrm{and}\;5\times 10}^{-3}$), $\eta_{e}$ of N=10 grows with time initially but reaches a plateau eventually. The time-invariant characteristic of $\eta_{e}$ of N=10 is also revealed by the $\varepsilon_{H}$-independent feature illustrated in the inset of Figure 1b. Similar to N=130, $\eta_{e}{(\varepsilon }_{H}{,\dot{\varepsilon }}_{H})$ decreases with increasing ${\dot{\varepsilon }}_{H}$ at a given $\varepsilon_{H}$. Note that the steady state $\eta_{e}$ is also observed for N=20, but a longer time period is required.

Although the stress and viscosity can be obtained in experiments, the microstructure evolution of polymer melts is difficult to observe. In contrast, the time-varying microstructure, such as polymer conformations subjected to stretching, can be captured by coarse-grained molecular simulations. To understand the microscopic evolution of polymer conformations during extension, the changes in the distribution of the radius of gyration with time *P*($R_{g},t$) are demonstrated in Figure 2. The average radius of gyration at equilibrium is ${\bar{R}}_{g0}$. Under stretching for N=130, the distribution becomes widened and shifts toward larger values of $R_{g}$, as shown in Figure 2a at the strain rate ${\dot{\varepsilon }}_{H}={5\times 10}^{-2}.$ Evidently, the increment of $R_{g}$ parallel to stretching (*x*-direction) is significantly more than the decrement of $R_{g}$ perpendicular to stretching (*y*- and *z*-direction). That is, the polymer conformation is elongated from a spherical shape. Two interesting findings are observed from *P*($R_{g},t$). It is found that the distribution *P*(t=0.16) before the onset of strain hardening is essentially the same as that at equilibrium. The deviation of *P*($R_{g},t$) from the equilibrium distribution becomes significant after strain hardening. Moreover, a bimodal distribution emerges after stretching for a long time (e.g., t=38.4 and 40), corresponding to large Hencky strains. One peak is close to $1.5{\bar{R}}_{g0}$, while the other peak is located at about 3.5 ${\bar{R}}_{g0}$. In addition to the Hencky strain, the bimodal distribution also depends on the strain rate. As depicted in the inset of Figure 2a for $\varepsilon_{H}=2$, the bimodal feature is more prominent for lower strain rates. The peak close to 2.5 ${\bar{R}}_{g0}$ is significantly higher than that close to 1.5 ${\bar{R}}_{g0}$. The appearance of the peak with a smaller radial size can be realized by the representative evolutions of polymer conformations shown in Appendix A. Some polymers are reoriented to the stretching direction with small elongation, some polymers are extended but then arrested in a specific stretched state, and some polymers return to the weakly stretched state via local relaxation. Appendix A depicts the typical evolutions of polymer conformations near the peak with a larger radial size. However, as the strain rate increases, the distribution of $R_{g}$ becomes more non-uniform because easily stretched polymers can be elongated further, and the number of polymers in a weakly stretched state decreases slightly.

In contrast to the unsteady stretching of long chains (N=130), steady stretching is reached for short chains (N=10). Figure 2b shows the time evolution of the distribution of $R_{g}$ for N=10 at ${\dot{\varepsilon }}_{H}={5\times 10}^{-2}$. By comparing Figure 2b with Figure 2a, it is found that the polymer size distributions of N=10 and 130 are distinctly different during stretching. While the bimodal distribution appears for long chains, the size distribution of the short chains is weakly perturbed by the equilibrium distribution. Moreover, *P*($R_{g},t$) eventually becomes time-invariant for N=10, but it still evolves toward the bimodal shape for N=130. The consequence of this is that subject to the same strain rate, the relaxation allows short chains to recover their shapes close to the equilibrium structure, but it fails to reduce the significant elongation of long chains. Furthermore, the effect of the strain rate on the size distribution of short chains at $\varepsilon_{H}=2$ is demonstrated in the inset of Figure 2b. At relatively low strain rates (${\dot{\varepsilon }}_{H}=10^{-2}{,5\times 10}^{-3}{,\;\mathrm{and}\;5\times 10}^{-4}$), it is found that their polymer size distributions are essentially the same as the equilibrium distribution. This result indicates that the relaxation is fast enough compared to the stretching for N=10. Their stretching dynamics always follow the linear viscoelastic response and ultimately reach the steady state. On the contrary, for high strain rates (${\dot{\varepsilon }}_{H}=10^{-1}{\;\mathrm{and}\;5\times 10}^{-2}$), the deviation from the equilibrium distribution is observed, revealing weak strain hardening, which cannot be clearly identified from $\eta_{e}\left(t\right)$.

### 3.2. Polymer Conformations Associated with Strain Hardening

The onset of strain hardening is generally identified as the point at which the elongational viscosity deviates from the linear viscoelastic response. However, $\eta_{0}\left(t\right)$ is difficult to acquire accurately because the strain rate needs to be as small as possible. At ${\dot{\varepsilon }}_{H}\to 0$, the thermal noise causes significant fluctuations of stress and $\eta_{e}\left(t\right)$ [25], as illustrated in Figure 1. It is desirable to understand the change in the microstructure of polymers near the onset point of strain hardening. Instead of the radial size distribution, the mean radius of gyration of polymers ${\bar{R}}_{g}$ is easier to use for the monitoring of the microstructural evolution during stretching. Figure 3a shows the variation of ${\bar{R}}_{g}{/\bar{R}}_{g0}$ with time at various strain rates for N=130. It is found that, regardless of ${\dot{\varepsilon }}_{H}$, all curves of ${\bar{R}}_{g}{/\bar{R}}_{g0}$ remain at unity for a while and then upturn rapidly at some points. The turning point occurs earlier when the strain rate is higher. The comparison of the onset points between Figure 1a and Figure 3a indicates that they coincided with each other for a given strain rate. This reveals that the upturn of the elongational viscosity is actually accompanied by the drastic change in the microstructure of the polymer melt. That is, strain hardening takes place as the average radial size of polymers starts to grow from the equilibrium state. Although the stretching process stops at $\varepsilon_{H}=2$ for all strain rates, the radial size of polymers is larger for higher ${\dot{\varepsilon }}_{H}$. To maintain a higher ${\dot{\varepsilon }}_{H}$, higher stress is required, which causes larger polymer deformation.

According to ${\bar{R}}_{g}{/\bar{R}}_{g0}$, the microstructure of the polymer melt is essentially the same as that at the equilibrium state before strain hardening. This result can be further illustrated by the plot of the standard deviation of ${\bar{R}}_{g}{/\bar{R}}_{g0}$ against time, as depicted in the inset of Figure 3a for N=130. Again, before the occurrence of strain hardening, the standard deviation is a constant and the same as that at equilibrium, σ=0.235. Similar to ${\bar{R}}_{g}{/\bar{R}}_{g0}$, the onset of strain hardening can be recognized at the point of the uprising of σ. The equilibrium state can be maintained for longer at lower strain rates. σ grows rapidly after the onset of strain hardening, signifying the incremental increase in the width of the radial size distribution. This occurs simply because the upper bound is elevated due to stretching. Since the higher stress for maintaining the higher ${\dot{\varepsilon }}_{H}$ yields more polymers with strong deformation, the distribution of $R_{g}{/\bar{R}}_{g0}$ is wider for the larger strain rate at $\varepsilon_{H}=2$.

According to *P*($R_{g},t$) and ${\bar{R}}_{g}\left(t\right)$, the linear viscoelastic response is observed as they are maintained at those of the equilibrium state. The onset of strain hardening corresponds to their deviation from the equilibrium state. Apparently, the initial elongational perturbation to the polymer conformation is not clearly displayed in $R_{g}$. To capture the response of the conformation to the stretching process, the aspect ratio (s) of the polymer is calculated. It is defined as the ratio of ${\bar{R}}_{\mathrm{gxx}}$ (parallel to the stretching direction) to ${(\bar{R}}_{\mathrm{gyy}}+{\bar{R}}_{\mathrm{gzz}})/2$ (perpendicular to the stretching direction) of the chain. Figure 3b shows the variation of the aspect ratio with time at different ${\dot{\varepsilon }}_{H}$ for N=130. Obviously, the aspect ratio is unity (s=1) at equilibrium, and it ascends rapidly at a certain point. At the end of the stretching ($\varepsilon_{H}=2$), ${\bar{R}}_{\mathrm{gxx}}$ can be as large as approximately eight to ten times (${\bar{R}}_{\mathrm{gyy}}+{\bar{R}}_{\mathrm{gzz}})/2$. The representative polymer conformations at different times are shown in the inset of Figure 3b, at ${\dot{\varepsilon }}_{H}={5\times 10}^{-3}$. Note that the significant deviation of the aspect ratio from unity can be identified before the onset of strain hardening. For example, one has ${\bar{R}}_{g}/{\bar{R}}_{g0}=1$ but s=1.1 at t=15.6 before strain hardening at ${\dot{\varepsilon }}_{H}={5\times 10}^{-3}$. This result reveals that the initial elongation tends to align the deformed polymers at the equilibrium state (with an aspect ratio greater than unity) with the stretching direction. That is, the equilibrium distribution *P*($R_{g},t$) is not altered actually but the mean orientation emerges because of the alignment induced by elongation.

It is imperative to realize the strain rate effect on strain hardening from the plot of ${\bar{R}}_{g}$ against the stretching time in terms of the characteristic onset time $t^{*}{(\dot{\varepsilon }}_{H})$. To understand the relationship between $t^{*}$ and ${\dot{\varepsilon }}_{H}$, the plot of ${\bar{R}}_{g}$ (t, ${\dot{\varepsilon }}_{H}$) can be redrawn against a dimensionless time ${\dot{\varepsilon }}_{H}t$, which is the Hencky strain $\varepsilon_{H}$. Figure 3c shows the variation of the mean radius of gyration with the elongation deformation, i.e., ${\bar{R}}_{g}/{\bar{R}}_{g0}$ vs. $\varepsilon_{H}$ for N = 130. For comparison, the plot of the standard deviation against $\varepsilon_{H}$ is depicted in the inset. It is somewhat surprising to find that the deviation of ${\bar{R}}_{g}/{\bar{R}}_{g0}$ from unity occurs essentially at the same Hencky strain, i.e., $\varepsilon_{H}\approx 0.1$, regardless of the strain rate. That is, despite differences in ${\dot{\varepsilon }}_{H}$, the onset of strain hardening always takes place at an approximately constant $\varepsilon_{H}$, which is consistent with the experiments [12,13]. In addition to the mean of the radial size distribution, similar behavior is observed for the variation of the standard deviation with $\varepsilon_{H}.$ The upturn of σ from the equilibrium value $\sigma_{0}$ takes place essentially at the same point ($\varepsilon_{H}\approx 0.1$) for all strain rates. This finding indicates that the linear viscoelasticity is always obeyed by very small Hencky strains, but the non-linear viscoelasticity emerges beyond the critical value ($\varepsilon_{H}>0.1$). Evidently, the radial size distribution *P* ($R_{g},t$) of the linear viscoelastic regime is independent of ${\dot{\varepsilon }}_{H}$, but that of the non-linear regime depends on ${\dot{\varepsilon }}_{H}$. Moreover, this finding also reveals that the characteristic onset time is inversely proportional to the strain rate, $t^{*}{\sim \dot{\varepsilon }}_{H}{}^{-1}$.

For long chains (N=130), strain hardening is easy to observe, but the steady elongational viscosity is difficult to reach. On the contrary, for short chains (N=10), the steady $\eta_{e}$ can be seen, but strain hardening cannot be recognized easily. Following the concept of Figure 3 for N=130, Figure 4 shows the time evolution of polymer conformations for N=10 at different strain rates. As demonstrated in Figure 4a, strain hardening can be clearly identified by ${\bar{R}}_{g}\left(t\right)$ at ${\dot{\varepsilon }}_{H}=10^{-1}{\;\mathrm{and}\;5\times 10}^{-2}$. However, at lower strain rates (${\dot{\varepsilon }}_{H}=10^{-2}{\;\mathrm{and}\;5\times 10}^{-3}$), the curve of ${\bar{R}}_{g}\left(t\right)$ does upturn at some point (weak strain hardening) but arrive at a plateau eventually (steady $\eta_{e}$). At the lowest strain rate (${\dot{\varepsilon }}_{H}={5\times 10}^{-4}$), the ${\bar{R}}_{g}\left(t\right)$ curve remains at unity all the time, indicating that the process follows the linear viscoelastic response and finally reaches the steady $\eta_{e}$. Note that the curve of $\eta_{e}\left(t\right)$ at ${\dot{\varepsilon }}_{H}={5\times 10}^{-4}$ is not shown in Figure 1b because it coincided closely with $\eta_{e}\left(t\right)$ at ${\dot{\varepsilon }}_{H}={5\times 10}^{-3}$.

Another dynamic feature of the radial size distribution *P*($R_{g}/{\bar{R}}_{g0},t$) is the standard deviation σ(t), which is illustrated in the inset of Figure 4a for N=10. It is found that the behavior of σ(t) of N=10 is distinctly different from that of N=130. After the onset of strain hardening for ${\dot{\varepsilon }}_{H}=10^{-1}{\;\mathrm{and}\;5\times 10}^{-2}$, the standard deviation σ(t) starts to grow fast because short chains are more easily stretched due to their weaker entanglement effects. Unlike σ(t) of N=130, the curves reach their peak and descend afterward, revealing that the relaxation of polymer conformations comes into play. Nonetheless, the steady state has not been reached, and hence the standard deviation still decays toward the equilibrium value. In contrast, at low strain rates (${\dot{\varepsilon }}_{H}\leq 10^{-2}$), σ(t) always fluctuates around the equilibrium value of 0.109 during the stretching process. This shows that the relaxation of polymer conformations dominates over the stretching of short chains at low strain rates.

The response of the polymer conformations along the stretching direction for N=10 is also demonstrated in terms of the aspect ratio (s) in Figure 4b. Again, a significant rise in the aspect ratio is accompanied by strain hardening at high strain rates. The value of s can be as high as about three. However, at lower strain rates, the rising aspect ratio reaches a plateau, which is consistent with the steady viscosity. The representative conformations at different times are depicted in the inset of Figure 4b at ${\dot{\varepsilon }}_{H}={5\times 10}^{-3}$. At the lowest strain rate, the value of s remains essentially at unity with a very slight deviation. This result discloses the fact that the equilibrium microstructure (radial size distribution) of short chains is always maintained at sufficiently low strain rates during the stretching process because of their rapid relaxation. Although the radial size distributions *P* ($R_{g},t$) are very similar at $\varepsilon_{H}=2$ for the lowest three strain rates (see the inset of Figure 2b), their aspect ratios can still be distinguished because the orientation effect is not revealed in *P*($R_{g},t$). The incremental increase in the elongational viscosity is attributed to the gradual alignment of deformed chains associated with the equilibrium state, which is achieved during stretching.

### 3.3. Effects of Molecular Weight

In addition to the strain rate, it is well-known that molecular weight is the property that affects the resistance to unentangling under strain as well [51]. That is, the elongational viscosity is expected to depend on the chain length, $\eta_{e}\left(N\right)$. In general, the melt strength of elongational viscosity is enhanced as the molecular weight increases [52]. Figure 5a shows the growth of $\eta_{e}$ with time at ${\dot{\varepsilon }}_{H}={5\times 10}^{-3}$ for various chain lengths, from N=30 to 390. The typical behavior of $\eta_{e}\left(t\right)$ is observed for each N, and the steady state is not reached. However, two regimes can be distinguished: the short and long time periods. At a short stretching time, all the elongational viscosities behave similarly regardless of the molecular weight. Moreover, they are essentially collapsed together even at different strain rates, as illustrated in the inset of Figure 5a. In contrast, at a sufficiently long time, $\eta_{e}\left(t\right)$ seems to separate from the main curve (asymptotic behavior) represented by the very long chain length, such as N=390. Obviously, the extent of the deviation from the main curve is more significant as the chain length is shorter. Because the linear viscoelastic response $\eta_{0}\left(t\right)$ varies with the molecular weight, it is difficult to identify the onset point of strain hardening for a given strain rate.

The characteristics of the two regimes displayed in $\eta_{e}\left(t\right)$ can be further understood by examining the radial size distribution. According to the identified onset point of strain hardening at ${\dot{\varepsilon }}_{H}={5\times 10}^{-3}$ for N=130, *P*($R_{g},t$) at t=20 (close to the onset point) remains the same as that at its equilibrium state. Figure 5b shows the radial size distributions at t=20 for N=30,60, and 260. Evidently, their *P*($R_{g}$) at t=20 still coincides with the corresponding equilibrium distributions. This outcome reveals that *P*($R_{g},t$) is unchanged in the short-time regime, and the onset point of strain hardening is not sensitive to the molecular weight. On the contrary, the radial size distribution deviates from the equilibrium distribution in the long-time regime. As shown in the inset of Figure 5b, the bimodal distribution appears at $\varepsilon_{H}=2$ only for N=60, but not for N=30 and 260. Similar to N=130 (see Figure 2a), the presence of the peak with a larger radial size implies that some polymers of N=60 have already elongated to their strongly stretched state. On the contrary, as the chain length becomes significantly longer than N = 130, most of the polymers of N = 260 fail to be extended to the strongly stretched state; thus, the bimodal distribution is not obvious. Certainly, for short chains (N = 30), the relaxation is much faster than that of N = 130 and *P*($R_{g},t$) shifts simply toward larger radial sizes upon stretching.

To understand the effect of molecular weight, the microscopic evolution of polymer conformations in terms of ${\bar{R}}_{g}\left(t\right)$ is illustrated in Figure 6a at ${\dot{\varepsilon }}_{H}={5\times 10}^{-3}$. In the initial period, all ${\bar{R}}_{g}{/\bar{R}}_{g0}$ remain in unity regardless of the molecular weight, indicating the regime of linear viscoelastic response. It is interesting to find that the upturn of all ${\bar{R}}_{g}/{\bar{R}}_{g0}$ curves seem to take place essentially at the same moment, revealing that the onset point of strain hardening is insensitive to molecular weight. Nonetheless, the growth rate of ${\bar{R}}_{g}$ varies with the chain length and ascends with increasing N. When N is large enough, the curve of ${\bar{R}}_{g}/{\bar{R}}_{g0}$ reaches an asymptotic behavior, similar to the behavior of $\eta_{e}/\eta_{0}$ (see Figure 5a). In addition to the mean (${\bar{R}}_{g}$), the characteristic of the radial size distribution can also be elucidated by the standard deviation σ. The inset of Figure 6a illustrates the variation of ${\sigma /\sigma }_{0}$ over time for different molecular weights, where $\sigma_{0}$ is the standard deviation of *P*($R_{g},t$) at the equilibrium state. Again, the onset of strain hardening (upturn from unity) seems to occur at about the same time. A maximum is observed for short chains (N = 30 and 60) due to their fast relaxation. The asymptotic behavior of σ(t) for long chains is observed as well.

The influence of the molecular weight on polymer conformations upon stretching can be illustrated by the aspect ratio (*s*) as well. The analysis of the strain rate effect indicates that the aspect ratio is more sensitive to the stretching process than the radius of gyration. In fact, the rapid growth of the aspect ratio occurs earlier than that of $\eta_{e}$ and ${\bar{R}}_{g}$, corresponding to the transition from the orientation to the stretching of polymers. Figure 6b shows the variation of *s* over time at ${\dot{\varepsilon }}_{H}={5\times 10}^{-3}$ for different molecular weights. For comparison, the plot of the radius of gyration along the stretching direction (${\bar{R}}_{\mathrm{gxx}}/{\bar{R}}_{\mathrm{gxx}0}$) against time is depicted in the inset. The rapid growth behavior of all of the curves seems to happen at about the same time for both *s*(t) and ${\bar{R}}_{\mathrm{gxx}}\left(t\right)$, regardless of the molecular weight. Moreover, sufficiently long chains behave alike, and their curves coincide with each other, revealing the asymptotic behavior of large N. According to ${\bar{R}}_{g}\left(t\right)$, σ(t), and *s*(t), at a given strain rate, the onset of strain hardening is insensitive to the molecular weight but very sensitive to the strain rate. For sufficiently long chains, the local environments (entanglement networks) look alike and cannot be distinguished easily. Therefore, the onset of strain hardening (weakly stretched) and the non-linear viscoelastic response (intermediately stretched) are similar. However, the degree of stretching, which varies with the molecular weight, is gradually revealed as the process continues. As a result, the onset of strain hardening collapses at short times, while the deviation from the asymptotic behavior of infinitely long chains appears at long times.

## 4. Conclusions

In this work, the microstructural evolutions of the linear polymer melt during the stretching processes are explored by dissipative particle dynamics. Consistent with experiments, the elongational viscosity $\eta_{e}\left(t\right)$ at different ${\dot{\varepsilon }}_{H}$ initially overlaps with the linear response $\eta_{0}\left(t\right)$, but it upturns at a shorter stretching time for a higher ${\dot{\varepsilon }}_{H}$. The onset of strain hardening is microscopically examined by the radial size distribution *P*($R_{g},t$). It is intuitively expected that *P*($R_{g},t$) grows widely and shifts toward larger values of $R_{g}$ upon stretching. However, it is interesting to find that *P*($R_{g},t$) remains essentially the same as that at equilibrium before strain hardening but is strongly deformed after that. A bimodal distribution can emerge after prolonged stretching, suggesting that not all the polymers are elongated. Some polymers have relaxed back into a weakly stretched state, while some are arrested in a specific stretched state. In contrast to the unsteady stretching of long chains, the steady elongational viscosity can be acquired for short chains. The *P*($R_{g},t$) of short chains is weakly perturbed from their equilibrium distribution, revealing that the relaxation is fast enough to recover the near-equilibrium conformations. Nonetheless, for a sufficiently small ${\dot{\varepsilon }}_{H}$, *P*($R_{g},t$) is always the same as the equilibrium distribution, indicating the absence of strain hardening.

The dynamics of the size distribution can be handily captured by the mean ${\bar{R}}_{g}\left(t\right)$ and standard deviation σ(t). The upturns of both ${\bar{R}}_{g}$ and σ from their equilibrium values signify the onset of strain hardening. However, the deviation of the aspect ratio ${\bar{R}}_{\mathrm{gxx}}{/[(\bar{R}}_{\mathrm{gyy}}{+\bar{R}}_{\mathrm{gzz}})/2]$ from unity is observed before strain hardening, indicating that the initial elongation tends to align the slightly stretched polymers with the stretching direction without altering the equilibrium distribution. In other words, the increment of the elongational viscosity associated with the linear response corresponds to the alignment of stretched polymers by maintaining the equilibrium size distribution. The strain rate dependence of strain hardening can be realized as ${\bar{R}}_{g}/{\bar{R}}_{g0}$, and ${\sigma /\sigma }_{0}$ are plotted against the dimensionless time ${\dot{\varepsilon }}_{H}t$. The close coincidence of the onset points suggests that the linear viscoelasticity always takes place at a sufficiently small Hencky strain. In addition to ${\dot{\varepsilon }}_{H}$, the viscosity depends on the chain length N, but the $\eta_{e}\left(t\right)$ of different N behave similarly in the initial period. On the basis of ${\bar{R}}_{g}/{\bar{R}}_{g0}$ and ${\sigma /\sigma }_{0}$, it is somewhat surprising to find that the onset of strain hardening is insensitive to the chain length. This is because the local environments (entanglement network) look alike for sufficiently long chains, and the weakly stretched state associated with the onset of strain hardening cannot be distinguished easily. Our study of the microstructure dynamics of polymer melt during stretching can serve as a crucial guide for mastering the art of polymer processing.

## Figures and Tables

**Figure 1 polymers-15-02067-f001:**
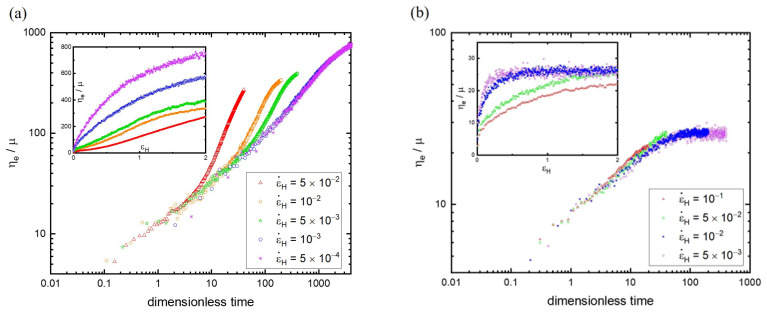
The time-varying elongational viscosities ($\eta_{e}$) for linear polymers with N=130 (**a**) and N = 10 (**b**) subject to various strain rates (${\dot{\varepsilon }}_{H}$). For the purpose of comparison, the elongational viscosity is non-dimensionalized by the shear viscosity of monomers. In the inset, the transient elongational viscosity is plotted against the Hencky strain ($\varepsilon_{H}$).

**Figure 2 polymers-15-02067-f002:**
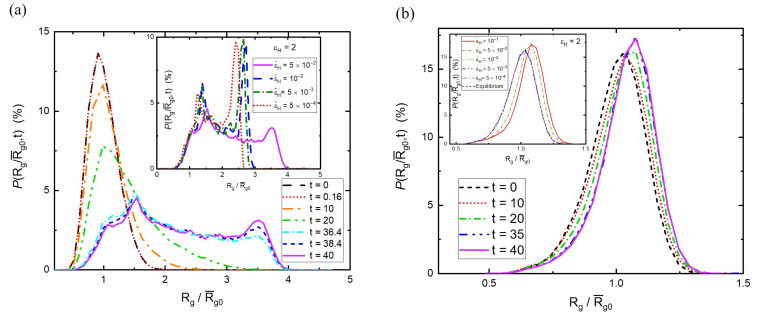
The time evolution of the distribution of $R_{g}$ under stretching for N=130 (**a**) and N = 10 (**b**) at the strain rate ${\dot{\varepsilon }}_{H}={5\times 10}^{-2}$. The inset demonstrates the effect of the strain rate on the size distribution at $\varepsilon_{H}=2$.

**Figure 3 polymers-15-02067-f003:**
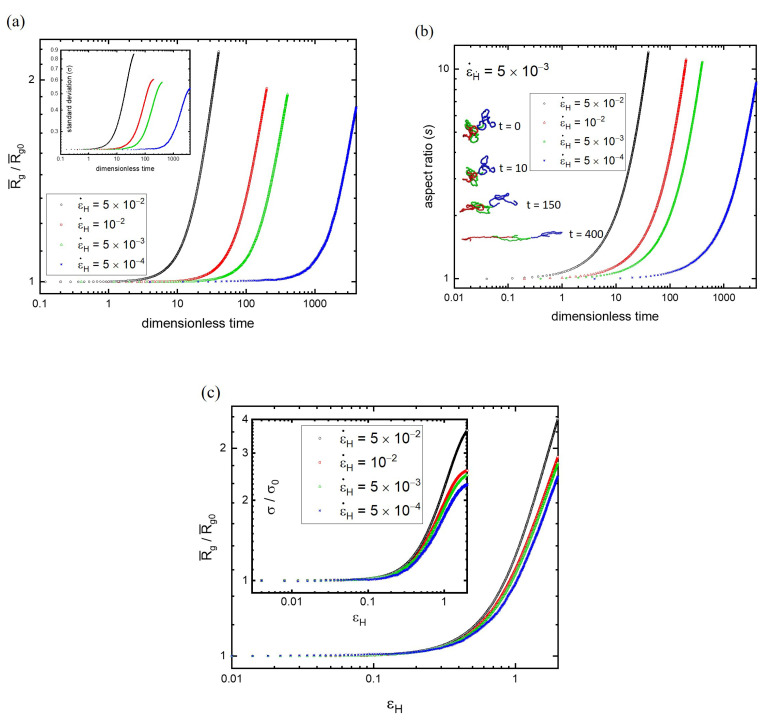
(**a**) The variation of ${\bar{R}}_{g}{/\bar{R}}_{g0}$ with time at various strain rates for N=130, where ${\bar{R}}_{g}$ is the mean radius of gyration. In the inset, the standard deviation (σ) of ${\bar{R}}_{g}{/\bar{R}}_{g0}$ is plotted against time. (**b**) The variation of the aspect ratio (*s*) with time at different $\dot{\varepsilon }$ for N=130. The representative polymer conformations at different times are shown in the inset at ${\dot{\varepsilon }}_{H}={5\times 10}^{-3}$. (**c**) The variation of the mean radius of gyration with the elongation deformation (${\bar{R}}_{g}/{\bar{R}}_{g0}$ vs. $\varepsilon_{H}$) for N = 130 at different ${\dot{\varepsilon }}_{H}$. For comparison, the plot of the standard deviation against $\varepsilon_{H}$ is depicted in the inset.

**Figure 4 polymers-15-02067-f004:**
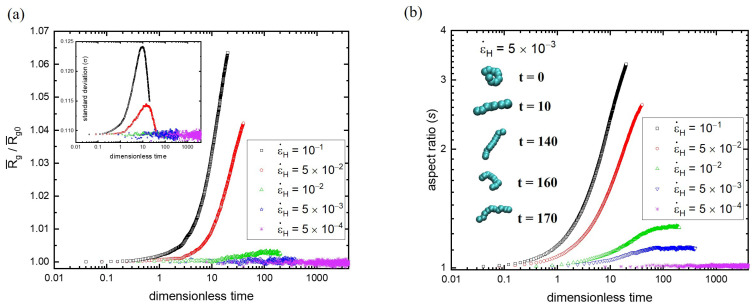
(**a**) The time evolution of polymer conformations for N=10 at different strain rates. In the inset, the standard deviation (σ) of ${\bar{R}}_{g}{/\bar{R}}_{g0}$ is illustrated. (**b**) The response of the polymer conformations along the stretching direction in terms of the aspect ratio (s) for N=10. The inset depicts the representative conformational evolution at ${\dot{\varepsilon }}_{H}={5\times 10}^{-3}$.

**Figure 5 polymers-15-02067-f005:**
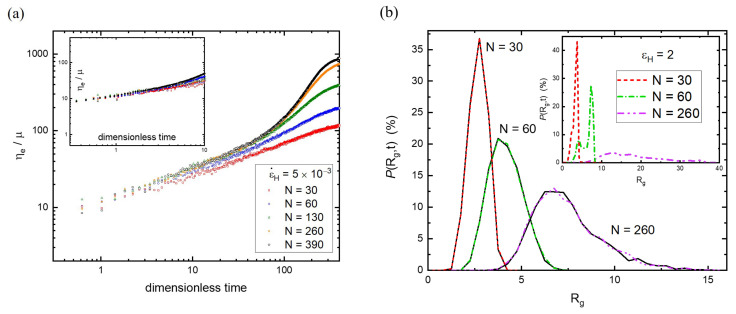
(**a**) The growth of $\eta_{e}$ with time at ${\dot{\varepsilon }}_{H}={5\;\times \;10}^{-3}$ for various chain lengths (N=30~390). In the inset, ${\dot{\varepsilon }}_{H}={5\;\times \;10}^{-2}$ is considered as well. The filled and hollow symbols represent ${\dot{\varepsilon }}_{H}={5\;\times \;10}^{-2}$ and ${\dot{\varepsilon }}_{H}={5\;\times \;10}^{-3}$, respectively. (**b**) The radial size distributions at t=0 and 20 for N=30, 60, and 260 at ${\dot{\varepsilon }}_{H}={5\;\times \;10}^{-3}$. The black solid line represents the size distribution at equilibrium. The radial size distributions at $\varepsilon_{H}=2$ are demonstrated in the inset.

**Figure 6 polymers-15-02067-f006:**
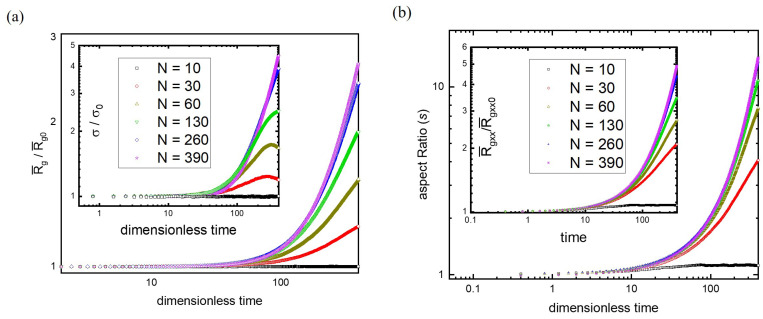
(**a**) The microscopic evolution of polymer conformations in terms of ${\bar{R}}_{g}\left(t\right)$ at ${\dot{\varepsilon }}_{H}={5\times 10}^{-3}$ for various molecular weights. The inset illustrates the variation of ${\sigma /\sigma }_{0}$ over time, where $\sigma_{0}$ is the standard deviation of *P*($R_{g},t$) at the equilibrium state. (**b**) The variation of the aspect ratio with time at ${\dot{\varepsilon }}_{H}={5\times 10}^{-3}$ for different molecular weights. For comparison, the plot of the radius of gyration along the stretching direction (${\bar{R}}_{\mathrm{gxx}}$/${\bar{R}}_{\mathrm{gxx}0}$) against time is depicted in the inset.

## Data Availability

The data presented in this study are available on request from the corresponding authors.

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
