# Peer review of "Microstructural Dynamics of Polymer Melts during Stretching: Radial Size Distribution"

_polymers, 2023, doi:10.3390/polym15092067_

Round 1

Reviewer 1 Report

In their manuscript entitled " Microstructural dynamics of polymer melts during stretching:
radial size distribution," Hsieh et al. investigated the radial size distribution of linear polymers during the stretching process by dissipative particle dynamics. The author found that both mean radius of gyration and standard deviation rise rapidly in the non-linear regime and the onset of strain hardening is insensitive to polymer chain length and Hencky strain.

The manuscript shows good quality and is well written, supported by simulation data. However, the authors should consider the following:

1.    The author should review more simulation papers for state-of-art polymer melt rheology. The author uses three paragraphs to introduce definitions like elongation viscosity, onset strain hardening and something else which might be tedious and hard to follow.

2.    Figures 1, 3, 4, 5 and 6 miss the unit of time.

3.    The author runs the simulation under stretching process. In the real process, however, polymers are normally treated with extruders. Will the results show a similar trend in the shearing process? If not, could the author explain why it is different?

Reviewer 2 Report

In this work, Tsao et al investigate time-resolved elongation viscosity of polymer melt using DPD simulation, and relates the polymer microstructures to the viscosity hardening behavior.   

The paper is well written, the simulation method is reasonable and can qualitatively reproduce experimental results. Although all the results are intuitive and expected, they are well presented and the discussions are in detailed; the findings of Rg stays close to equilibrium before onset of viscosity hardening is centrally informative.  Given the completeness of the paper, I only have minor comments for revision: 

1) If I understand correctly, the simulation needs to reach equilibrium before performing elongation study, can the author clarify and show evidence that their systems reach equilibrium? 

2) Lines 170~182 are kind of repetitive from the introduction. I suggest the authors replace it with one or two concise sentences
